# TbsNet: the importance of thin-branch structures in CNNs

Xiujian Hu[1], Guanglei Sheng[2], Piao Shi[3] and Yuanyuan Ding[1]

[1] Department of Electronics and Information Engineering, Bozhou University, Bozhou, AnHui, China
[2] School of Computer Science and Engineering, Xi'an University of Technology, Xi'an, Shaanxi, China
[3] School of Artificial Intelligence, Hefei University of Technology, Hefei, Anhui, China



## ABSTRACT

The performance of a convolutional neural network (CNN) model is influenced by several factors, such as depth, width, network structure, size of the receptive field, and feature map scaling. The optimization of the best combination of these factors poses as the main difficulty in designing a viable architecture. This article presents an analysis of key factors influencing network performance, offers several strategies for constructing an efficient convolutional network, and introduces a novel architecture named TbsNet (thin-branch structure network). In order to minimize computation costs and feature redundancy, lightweight operators such as asymmetric convolution, pointwise convolution, depthwise convolution, and group convolution are implemented to further reduce the network's weight. Unlike previous studies, the TbsNet architecture design rejects the reparameterization method and adopts a plain, simplified structure which eliminates extraneous branches. We conduct extensive experiments, including network depth, width, *etc*. TbsNet performs well on benchmark platforms, Top 1 Accuracy on CIFAR-10 is 97.02%, on CIFAR-100 is 83.56%, and on ImageNet-1K is 86.17%. Tbs-UNet's DSC on the Synapse dataset is 78.39%, higher than TransUNet's 0.91%. TbsNet can be competent for some downstream tasks in computer vision, such as medical image segmentation, and thus is competitive with prior state-of-the-art deep networks such as ResNet, ResNeXt, RepVgg, ParNet, ConvNeXt, and MobileNet.

## INTRODUCTION

Convolutional neural networks (CNNs) can express highly complex feature information and have shown great success in solving many classical computer vision problems, such as image classification, detection, and segmentation (*He et al., 2016a; Long, Shelhamer & Darrell, 2015; Redmon & Farhadi, 2018; Ronneberger, Fischer & Brox, 2015*). The depth and width of the network, the size of the receptive field, the size of the feature map, the number of parameters (params), and the amount of floating-point computation (FLOPs) have many effects on the performance of the convolutional neural network (*Ding et al., 2021; Liu et al., 2022a; Wightman, Touvron & Jégou, 2021*). To reduce the number of parameters and decrease the cost of floating-point computations, lightweight operators

Corresponding author
Xiujian Hu,
2015020002@bzuu.edu.cn

represented by pointwise (PW) convolution, depthwise (DW) convolution, group convolution, and asymmetric convolution, *etc.*, have been widely studied and applied, such as Inception (*Liu et al., 2021a*), GhostNets (*Han et al., 2022*), ACNet (*Ding et al., 2019*), RepLKNet (*Ding et al., 2022*), ResNet (*He et al., 2016a*), MobileNets (*Sandler et al., 2018*) *etc*. In addition, multi-branch network structure with the ability of multi-scale feature extraction has been the focus of research, such as RepVgg (*Ding et al., 2021*), SegNeXt (*Guo et al., 2022*), ConvNeXts (*Liu et al., 2022a*), ParNet (*Goyal et al., 2021*), *etc*. However, the multi-branch structure has the disadvantages of long inference time and high resource occupancy, so it has to adopt the reparameterization method to design the inference model into an efficient plain structure, such as RepVgg (*Ding et al., 2021*), ParNet (*Goyal et al., 2021*), *etc*.

How to optimize the combination of various influencing factors is the main difficulty in network model design (*Tan & Le, 2019*, *2021*; *Wightman, Touvron & Jégou, 2021*). (1) The tradeoff between depth and width. Expanding the width can improve the performance but increase the computation cost highly. Although incrementing the depth of the network can enhance the performance at fewer costs, it will also bring some problems, such as increasing the difficulties in training (*He et al., 2016b*). In the case of both increasing depth and expanding width, the best combination of depth and width needs to be studied to reduce the params and FLOPs (*Zhang & Ding, 2023*). (2) The design of network structure. Although the multi-branch structure has the advantages of multi-scale feature extraction, it also has disadvantages, such as long inference time and more resource cost. The plain structure has faster inferring but no multi-scale feature extraction (*Goyal et al., 2021*). Although parallel architecture is efficient and fast, its application needs the support of a hardware platform. (3) The contradiction between receptive field size and lightweight. Increasing the receptive field size can obtain more advanced features, and the simple method is to enlarge the size of convolution kernels (*Zhang et al., 2022*). However, the use of large convolution kernels increases the params and FLOPs of the network (*Ding et al., 2022*; *Liu et al., 2022a*). (4) The tradeoff between feature map scaling and computing performance. The larger size of feature maps can bring richer feature information, but the more computation cost will also be, which may make the model untrainable (*Redmon & Farhadi, 2018*).

We study the key factors such as the depth, width, receptive field size, lightweight operator, branching structure, and feature map size, gain trade-offs in terms of depth and width of the network, and design a convolutional neural network named TbsNet (thin-branch structure network) with the simplified multi-branch structure to keep the network plain as a whole. Our contributions are mainly about three aspects as follows.

- The TBSBlock based on a thin-branch structure is constructed. TBSBlock has the advantages of both muti-branch and plain models and does not need reparameterization.
- We propose strategies for how constructing an efficient CNN architecture. TbsNet has good scalability, lightweight and easy training, and skillfully avoids the emergence of network degradation caused by excessive depth.

- TbsNet optimizes the combination of the factors and performs well on benchmark platforms such as CIFAR-10, CIFAR-100, Tiny-ImageNet, ImageNet-1K, *etc*. TbsNet can be competent for some downstream tasks in computer vision, such as medical image segmentation. Tbs-UNet's DSC is higher than TransUNet's 0.91 percentage points, and HD is lower than 0.56 (mm) on the Synapse dataset (*Chen et al., 2021*).

## RELATED WORKS

The impact of depth and width on the network performance is shown in this section, the relationship between kernel size and receptive field is analyzed, and the advantages and disadvantages of multi-branch are enumerated. We also propose strategies to construct an efficient CNN architecture.

### The depth and width

*Cybenko (1992)* proposed that a single-layer neural network with Sigmoid activation can approximately represent any function with arbitrarily small error on the condition that the network must have enough width, which may increase the params and FLOPs greatly. *Liang & Srikant (2016)* proposed that the non-deep network needs more exponential neurons than the deep network to achieve the accuracy of function approximation, which is one of the main advantages of deep neural networks. With the emergence of ResNet (*He et al., 2016b*) with up to 1,000 layers, increasing the depth of the network to improve network performance has become a consensus. Surprisingly, a Transformer with more than 1,000 layers has also been designed and performs training stably (*Wang et al., 2022*). However, the shallow network ParNet (*Goyal et al., 2021*) with only 12 layers has achieved success by increasing the width on a large scale, which has once again aroused the discussion of the network structure in the artificial intelligence community.

The depth and width of the network play an important part in scaling the size of params and FLOPs. For any input Features = (B, $C_{in}$, H, W), the shape of the convolution kernel tensor is Kernel = (($k_h$, $k_w$), S, $C_{in}$, $C_{out}$), where B is the size of batch size, H and W are the height and width of the input feature map respectively, ($k_h$, $k_w$) represents the height and width of the convolution kernel respectively, S is the step size (stride), $C_{in}$ and $C_{out}$ represent the number of input channels and output channels respectively. $C_{in} \times k_h \times k_w$.

### *The params*

The params for each convolution layer are calculated as follows,

$$Params = C_{out} \times (k_h \times k_w \times C_{in} + 1) \tag{1}$$

In Formula (1), $C_{in} \times k_h \times k_w$ represents the weight number of a filter, '+1' means adding the weight number of a bias, and $C_{out}$ represents the number of filters in the layer. If $k_h = k_w = k$,

$$Params = C_{out} \times (k^2 \times C_{in} + 1) \tag{2}$$

with the batch normalization (BN) operator used, the model does not need a bias, and then '+1' is removed from Formulas (1) and (2).

### The FLOPs

The FLOPs of each convolution layer are related not only to the params but also to the size of the input feature map. The calculation of FLOPs is as follows,

$$FLOPs = [(C_{in} \times k_h \times k_w) + (C_{in} \times k_h \times k_w - 1) + 1] \times C_{out} \times H \times W \qquad (3)$$

In Formula (3), $C_{in} \times k_h \times k_w$ represents the number of multiplications in a filter, $C_{in} \times k_h \times k_w - 1$ represents the number of additions in a filter, '+1' means adding the weight number of a bias, $C_{out} \times H \times W$ represents the number of output features. If $k_h = k_w = k$, the Formula (3) could be simplified as follows,

$$FLOPs = 2 \times C_{in} \times k^2 \times C_{out} \times H \times W \qquad (4)$$

If a 'Multi-Add' (a combination of addition and multiplication) operation is treated as a single floating-point calculation, Formula (4) can be further simplified as follows,

$$FLOPs = C_{in} \times k^2 \times C_{out} \times H \times W \qquad (5)$$

It can be seen from Formulas (2) and (5) that the params and FLOPs of the network have a first-order linear relationship with the changes in the number of channels and a second-order polynomial relationship with the changes in the kernel size. The above result is the same for all the layers in a network, which means that the depth and width are all in a linear relationship with params and FLOPs.

## Kernel size and receptive field

The receptive field has a powerful influence on the performance of CNN. The success of Transformer in computer vision has inspired the artificial intelligence community to re-examine the shortcomings of convolutional neural networks (*Dosovitskiy et al., 2021*; *Han et al., 2023*; *Lin et al., 2022*; *Liu et al., 2021b*, *2022a*). The self-attention mechanism of the Transformer has the ability of global perception, which exceeds the receptive field determined by the kernel size, which becomes one of the main gaps between the Transformer and CNN (*Han et al., 2023*). The remarkable advantage of the convolutional operator lies in its outstanding local perception ability, which uses a large convolution kernel to obtain a larger local receptive field, thus narrowing the gap between the convolutional operator and Transformer (*Ding et al., 2022*; *Liu et al., 2022a*).

The receptive field represents the range region of a specific CNN feature in the input space, and we can calculate the size by two factors (the center position and the feature map size) (*Dang Ha The Hien, 2017*). The calculation of receptive field size involves feature numbers (F) in each dimension, the size of the current receptive field (R), the distance between two adjacent features (J), and the center coordinate of the features (Cen). And Cen is taken to be the central coordinate of the receptive field. For any input feature $F_{in}$, the kernel size is k, the step size is s, the padding size is p, and the output feature is $F_{out}$,

$$F_{out} = \left\lfloor \frac{F_{in} + 2p - k}{s} \right\rfloor + 1 \tag{6}$$

The distance between two adjacent features in the output feature map ($J_{out}$) is equal to the product of the distance step size between them,

$$J_{out} = J_{in} \times s \tag{7}$$

The receptive field size $RF_{out}$ is calculated by the formula as follows,

$$RF_{out} = RF_{in} + (k - 1) \times J_{in} \tag{8}$$

And the center position of the receptive field of the output features is calculated by the formula as follows,

$$Cen_{out} = Cen_{in} + \left( \frac{k - 1}{2} - p \right) \times J_{in} \tag{9}$$

Formula (8) shows that the receptive field size is closely related to the kernel size. The kernel sizes commonly used are 3×3, 5×5, 7×7, and more large convolution kernels with the size of 9×9, 13×13, 31×31, *etc.* However, Formulas (2) and (5) also show that kernel size has a powerful impact on params and FLOPs. Even if large and super-large kernel sizes are more effective in some network structures, more applications of a large kernel are bound to cause more computation costs and make the network untrainable. An ingenious alternative is to replace the large kernel with multiple stacked Conv3×3 to obtain the equivalent receptive field size, which has a better performance in terms of computation, such as replacing a Conv5×5 with two Conv3×3 (*Ioffe & Szegedy, 2015*) and replacing a Conv7×7 with three Conv3×3 (*Liu et al., 2021c, 2022a*; *Simonyan & Zisserman, 2015*). There is a tradition of stacked Conv3×3 in classical networks, such as Vgg (*Simonyan & Zisserman, 2015*), ResNets (*He et al., 2016a*), ResNeXt (*Xie et al., 2017*), *etc.*

*Luo et al. (2016)* proposed the effective receptive field (ERF) theory at first. *Luo et al. (2016)* found that ERF is related to the theoretical receptive field, but not all pixels in the receptive field have the same contribution to the output vector. In many cases, the influence of pixels in the receptive field obeys the Gaussian distribution, the ERF is only part of the theoretical receptive field, and the Gaussian distribution attenuates rapidly from the center to the edge. In addition, the size of the ERF is also affected by the network structure. For example, although the residual network structure solves the problem of network degradation well, the Identity Map makes the ERF smaller (*Luo et al., 2016*).

## Multi-branch structure

Researchers have never stopped making a full study of the network structure to improve network performance. The plain structure has the advantages of fast inferring and common cost, while the multi-branch structure can obtain multi-scale feature information. RepVgg (*Ding et al., 2021*) uses the multi-branch structure to improve the classical Vgg (*Simonyan & Zisserman, 2015*) network and achieves good performance. ParNet (*Goyal*

*et al., 2021*) uses the double-branch parallel structure to verify that the shallow network can also play an excellent performance, and can greatly reduce the number of network parameters. Inception is a relatively early successful multi-branch structure, and the residual structure in ResNet (*He et al., 2016a*) is also a branch structure in nature.

However, adopting more branches in the network will harm performance (*Cheng et al., 2020*; *Xu & McAuley, 2023*). The results are mainly about two aspects. (1) The params are too large, and it is easy to produce an overfitting phenomenon. (2) It takes up more memory cost and inference time, which is not conducive to training the model.

As one of the optimization methods of multi-branch structure, reparameterization has attracted more and more attention in recent years. RepVgg (*Ding et al., 2021*) adopts the reparameterization method to speed up the inference speed. However, the inference model structure needs to be redesigned when using the reparameterization method, so reparameterization is limited and is not suitable for all architectures.

## Strategies for constructing CNNs

In summary, we propose several strategies for convolutional neural network design. (1) Network scaling in three dimensions: depth, width, and feature graph size. Scaling the depth and width of the network has a significant impact on FLOPs and params, while choosing the right depth and width can improve network performance without significantly increasing computing costs. In addition, the size of the input feature map also has an important effect on the network performance. (2) To extract multi-scale feature information, branch structure is indispensable. In addition to multi-scale feature sampling, multi-branch structure also has feature alignment function during feature fusion. (3) The overall structure of the network should be kept plain to reduce feature redundancy and inference time. A multi-branch structure increases memory occupancy and inference time, while a plain structure has faster inference time and less memory search. (4) Lightweight operators such as pointwise convolution, depthwise convolution, asymmetric convolution, and group convolution can be adopted to reduce the params and FLOPs of the network. (5) Network design must pay attention to the size of the effective receptive field. The branch structure has a certain effect on the size of the effective receptive field. (6) The reparameterization method is not necessary for any architecture. The goal of the reparameterization method is to reparameterize the learning weight of the multi-branch structure network model into the weight of a plain structure used, but not every network model can be reparameterized.

## METHODS

In this section, a perfect tradeoff will be gained among the depth, width, receptive field size, feature graph size, params, and FLOPs of the network TbsNet we designed, which has a plain structure as a whole. Our designed TbsBlock is the basic module of TbsNet. Compared with other branch structures, TbsBlock has the least number of branches, this enables TbsNet to have a simple structure and fast inference speed. Compared with ParNet, we maintain the necessary feature information and effectively reduce redundancy only through the simplified branch structure, without using the attention mechanism and

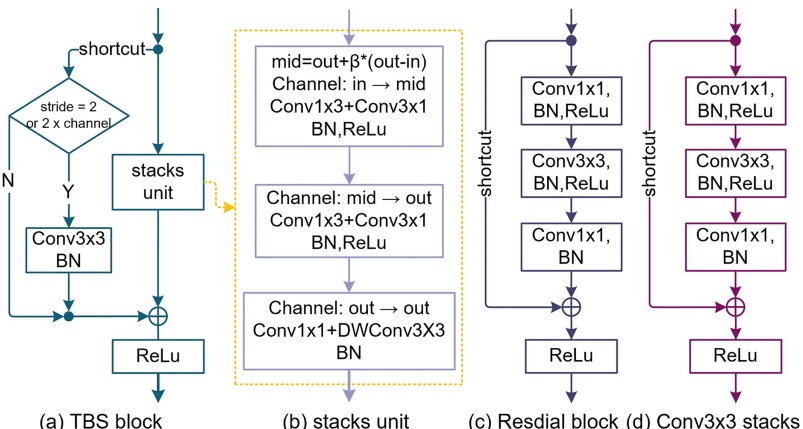

(a) TBS block    (b) stacks unit    (c) Resdial block (d) Conv3x3 stacks

**Figure 1   The design of the TBS block (thin-branch structure block).** (A) Is the TBS block we designed. (B) Is the stack unit of the TBS block which is based on pointwise Conv1×1, depthwise Conv3×3, SE (squeeze and expand), and Asymmetric Conv. (C) Is the bottleneck block of ResNets. (D) Is a simple stacking of Conv3×3 to elevate the receptive field.     

reparameterization method. The depth of the network model is at least 17 layers and up to 44 layers, and the downsampling ranges from 8X to 32X, which properly avoids the problem of network degradation and reduces the difficulty of network training.

## TBS block

Although the residual structure (*He et al., 2016b*) solves the problem of network degradation very well, the Identity map makes the effective perception field smaller (*Luo et al., 2016*). To overcome the problem of receptive field degradation, TbsNet draws lessons from the success of Residual block and Inception in structure, adopts a double-branch structure similar to Residual block, studies and improves the Identity map and Bottleneck units in Residual block (Fig. 1).

### Improvements on the shortcut

In the residual block of ResNet (*He et al., 2016b*, *2016a*), the function of the shortcut is identity mapping. We improve the shortcut branch and try to replace Conv1×1 with Conv3×3 to obtain richer advanced feature information and make it have better feature alignment ability (Fig. 1). Compared with the Residual block (Fig. 1), the function of the shortcut is aligning features rather than preventing network degradation.

### Design of stack unit

In Fig. 1, we replace a Bottleneck module by stacking three Conv3×3 to improve the receptive field. This method is similar to the reparameterization of RepVgg (*Ding et al., 2021*), with the drawback of increasing params and FLOPs (Fig. 1D). To reduce the params, FLOPs, and redundant features of the model, the first and second Conv3×3 were replaced by asymmetric convolution (Conv1×3 + Conv3×1), and the third by pointwise convolution and depthwise convolution (Conv1×1 + Conv3×3) (Fig. 1B). The TBS block overall maintained the receptive field size of three Conv3×3 stacks.

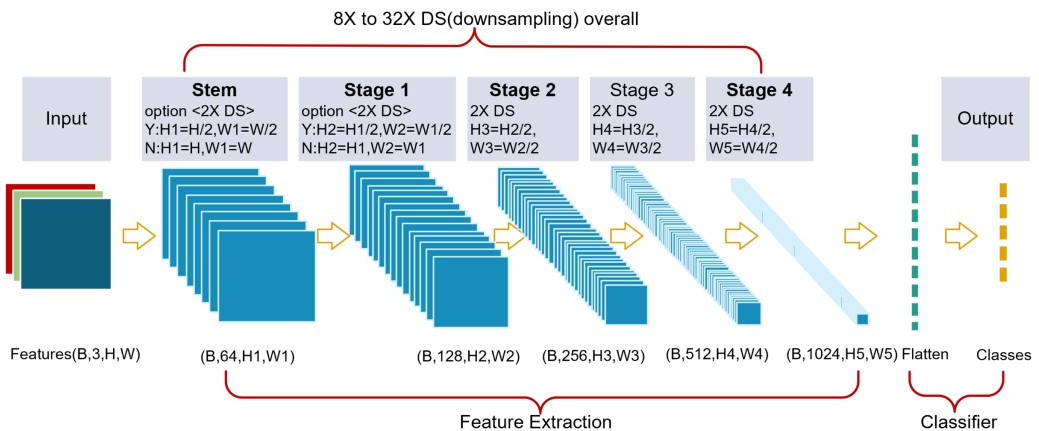

**Figure 2  We designed a thin-branch structure network.** The network is a plain structure, and the last block from Stage 2 to Stage 4 performs 2X DS (downsampling). The Stem block and Stage 1 keep 2X downsampling optional.

*Feature fusion*

TBS block abandons the channel attention and spatial attention mechanism, uses the expansion and squeeze (ES) method to fuse the channel information multiple times, and sets the parameter mid as the number of temporary channels.

$$mid = out + \beta * (out - in) \tag{10}$$

where "$\beta$" ($\beta \in \{-1, 1\}$) is the expansion coefficient; "out" is the number of output channels; "in" is the number of input channels. In Fig. 1B, the number of input channels of the first asymmetric convolution (Conv1×3 + Conv3×1) is "in" and the number of input channels is "mid". The input channel number of the second asymmetric convolution (Conv1×3 + Conv3×1) is "mid" and the input channel number is "out". The number of input channels and the number of output channels of the pointwise convolution (PWConv1×1) and the depthwise convolution (DWConv3×3) are both "out".

## Architecture

*Overall structure*

A thin-branch structure network (TbsNet) was constructed by stacking multiple TBS blocks (Fig. 2). TbsNet consists of one Stem module, four Stage modules, and one Classifier module, Stage = {$S_1$, $S_2$, $S_3$, $S_4$}. The Stem module contains only 1 Conv3×3+BN+ReLu, and its function is to expand the number of channels of the input feature map to 64, and whether to 2X downsampling is controlled by the step size. The Stage module consists of multiple TBS blocks, and only the last TBS block performs a judgment to expand the channels and 2X downsampling. The total number of TbsNet branches is four, which is much lower than other branch structure networks. The last TBS block of the $S_1$ module controls whether to 2X downsampling by the parameter stride, $S_2$, $S_3$, and $S_4$ complete 2X downsampling in the last TBS block. The Classifier module flattens the input feature information and outputs the result according to the number of classes.

**Table 1 Network width settings for every stage in TbsNet.**

| width$_{list}$ | S$_1$ | S$_2$ | S$_3$ | S$_4$ |
|---|---|---|---|---|
| $w_1$ | 96 | 192 | 384 | 512 |
| $w_2$ | 128 | 256 | 384 | 768 |
| $w_3$ | 128 | 256 | 512 | 1,024 |
| $w_4$ | 128 | 256 | 512 | 2,048 |

**Note:**
Stage = {S$_1$, S$_2$, S$_3$, S$_4$}, width$_{list}$ = {w$_1$, w$_2$, w$_3$, w$_4$}.

**Table 2 Variant structure and params, FLOPs.**

| Model alias | Depth (Layers) | Blocks/Stage | Width | ß | | Params (M) | | FLOPs (G) | |
|---|---|---|---|---|---|---|---|---|---|
| $N_1$ | 17 | [1, 1, 2, 1] | $w_1$ | −1 | 1 | 6.31 | 9.27 | 23.46 | 35.12 |
| $N_2$ | 23 | [1, 2, 3, 1] | $w_2$ | −1 | 1 | 10.27 | 16.97 | 56.03 | 76.61 |
| $N_3$ | 29 | [1, 3, 4, 1] | $w_3$ | −1 | 1 | 16.27 | 28.23 | 81.29 | 108.91 |
| $N_4$ | 35 | [1, 3, 6, 1] | $w_3$ | −1 | 1 | 17.9 | 29.83 | 101.34 | 128.96 |
| $N_5$ | 44 | [1, 3, 9, 1] | $w_4$ | −1 | 0 | 30.96 | 54.94 | 139.47 | 167.06 |

**Note:**
The basic settings of the TBS block are as follows. Shortcut = Conv3×3 and set_group = 0. Depth$_{list}$ = {17, 23, 29, 35, 44}.

TbsNet structure keeps plain on the whole, and the adjustable downsampling multiple can flexibly adapt to feature images with different resolutions, which enables TbsNet to have good scaling ability. Compared with ResNet (*He et al., 2016b*), 2X downsampling of TbsNet exists in the last TBS block, which could reduce the number of unnecessary branches.

Compared with RepVgg, Inception and other modules, we design modules with fewer branches, and compared with ResNet, our short-hop connection is downsampled only four times in four stages. The network architecture we designed is thin in general.

### Variants and settings

We set up five different depth values and four different width lists, depth$_{list}$ = {17, 23, 29, 35, 44}, width$_{list}$ = {w$_1$, w$_2$, w$_3$, w$_4$}, and different network variant structures are obtained by combining the depth and width. For example, "depth_id" = 2 means that the depth of the network is the value with index = 2 in the depth$_{list}$. "set_group" controls whether to set the group number for the asymmetric convolution in the TBS block (Fig. 1), and the stride is to set 2X downsampling in the Stem block and Stage 1. When "stride" = 1, it means no downsampling by a factor of 2, while "stride" = 2 means downsampling by a factor of 2. The parameter named expansion corresponds to the coefficient "β" (Formula (10)).

There are four lists in the width$_{list}$, each containing four values corresponding to the number of output channels of the four stages (Table 1). According to Formula (10), the expansion coefficient "β < 0" indicates that the width of the network is squeezed and then expanded, and "β > 0" denotes that expansion is first and squeezed second.

The five variant network structures are listed in Table 2 along with the corresponding params and FLOPs. For more variant structures, please refer to Appendix 1. The depth of the network is indicated by the number of convolution layers. Each TBS block of TbsNet

includes three convolution layers, Stem and classifier include one convolution layer on one's own, and then the total number of network layers is calculated as $depth_k = 2 + 3 \times \sum_{i=1}^{4} blocks_i, (k \in [1, 5])$. Parameters were calculated by torchsummary (https://github.com/tyleryep/torchsummary) and FLOPs were fvcore (https://github.com/facebookresearch/fvcore). The size of features determines the amount of computation. All FLOPs are calculated in the case of input feature = (3, 224, 224).

The depth of TbsNet ranges from 17 to 44, and the number of feature output channels is between 512 and 2,048. In the above five variant structures, we can see that the minimum of parameters is 6.31M and the maximum is 54.94M, to the FLOPs, the minimum FLOPs is 23.46G and the maximum FLOPS is 167.06G.

### Network lightweight

The overall architecture of TbsNet reduces unnecessary branches as much as possible to reduce feature redundancy and inference time. Lighted operators such as Pointwise Conv, Depthwise Conv, Asymmetric Conv, and Group Conv are used in the TBS block to reduce params and FLOPs. Specifically, three lightweight methods are adopted to decrease params and FLOPs. (1) Replace 2 Conv3×3 with an asymmetric convolution (Conv1×3 + Conv3×1). (2) Keep Conv1×1 as a shortcut. (3) Group the asymmetric convolutions. The setting of groups of the first and second Asymmetric Conv (Conv1×3 + Conv3×1) in the TBS block are 4 and 8, respectively. The network after lighted has smaller parameters and floating-point calculations, with a minimum params of 1.4M and minimum FLOPs of 5.53G.

## ABLATION STUDY

The experimental design in this section will verify the effectiveness of our designed TbsBlock and TbsNet architectures. Three performance indicators, params, FLOPs, and accuracy, are selected for comparison. Since the inference time of network architectures varies greatly under different hardware environments and software platforms, it is difficult to use inference time as a fair performance metric. However, params and FLOPs determine the required memory size and amount of computation, so we choose params, FLOPs and accuracy as performance indicators. The content of the experiment is mainly about three aspects. (1) The performance difference between Conv3×3 and Conv1×1 as a shortcut is compared. (2) The influence of Asymmetric Conv with groups on the network performance indicators. (3) The effect of network scaling on the network performance indicators. We abandoned any training techniques and only adopted the naive unified methods to train the network in experiments, without fining the model or optimizing the hyperparameters and using data augmentation methods mentioned in *Liu et al. (2022a)*, *Wightman, Touvron & Jégou (2021)*.

The experiment environment is Python3.8.6, Pytorch1.8.2, and Cuda11.3. All the experiments are running on one Nvidia 3090GPU. The benchmark test platform is CIFAR-10 and CIFAR-100 (*Krizhevsky, 2009*) datasets, and the global random seed is "seed = 1234". The training hyperparameter is set as follow, epoch = 200, bs = 128,

**Table 3 Comparison of performance indicators between Conv3×3 and Conv1×1.**

| Depth (Layers) | Width | Set group | β | Conv1×1 | | | Conv3×3 | | | Acc (↑↓) |
|---|---|---|---|---|---|---|---|---|---|---|
| | | | | Params (M) | FLOPs (G) | Acc (%) | Params (M) | FLOPs (G) | Acc (%) | |
| 17 | $w_1$ | × | 1 | 7.02 | 28.23 | 95.92 | 9.27 | 35.12 | 95.92 | 0.00 |
| | | √ | 1 | 1.91 | 7.54 | 95.26 | 4.16 | 14.43 | 95.66 | 0.40[†] |
| | $w_2$ | × | 1 | 12.64 | 46.34 | 96.02 | 15.95 | 56.49 | 96.48 | 0.46[†] |
| | | √ | 1 | 3.27 | 12.07 | 95.66 | 6.59 | 22.22 | 95.91 | 0.25[†] |
| | $w_3$ | × | 1 | 20.87 | 56.41 | 96.18 | 26.18 | 68.66 | 96.38 | 0.20[†] |
| | | √ | 1 | 5.32 | 14.49 | 95.81 | 10.63 | 26.74 | 96.14 | 0.33[†] |
| | $w_4$ | × | 0 | 39.51 | 61.38 | 95.7 | 48.82 | 76.70 | 96.11 | 0.41[†] |
| | | √ | 1 | 14.56 | 21.51 | 95.9 | 23.87 | 36.82 | 96.01 | 0.11[†] |
| 23 | $w_1$ | × | 1 | 7.59 | 39.59 | 96.48 | 9.84 | 46.48 | 96.31 | −0.17 |
| | | √ | 1 | 2.05 | 10.50 | 95.95 | 4.3 | 17.39 | 96.01 | 0.06[†] |
| | $w_2$ | × | 1 | 13.66 | 66.47 | 96.49 | 16.97 | 76.61 | 96.75 | 0.26[†] |
| | | √ | 1 | 3.53 | 17.26 | 96.19 | 6.85 | 27.41 | 96.21 | 0.02[†] |
| | $w_3$ | × | 1 | 21.89 | 76.53 | 96.4 | 27.21 | 88.78 | 96.68 | 0.28[†] |
| | | √ | 1 | 5.58 | 19.69 | 96.27 | 10.89 | 31.94 | 96.1 | −0.17 |
| | $w_4$ | × | 0 | 40.53 | 81.50 | 96.12 | 49.84 | 96.82 | 96.46 | 0.34[†] |
| | | √ | 1 | 14.82 | 26.70 | 96.13 | 24.13 | 42.01 | 96.31 | 0.18[†] |

**Note:**
[†] Denotes that the $Acc_{Conv3×3}$ is greater than $Acc_{Conv1×1}$. The "set_group" indicates whether to set the number of groups for the asymmetric convolution in the TBS block module and the default groups are four and eight. Dataset = CIFAR-10. Acc (↑↓) = $Acc_{Conv3×3} − Acc_{Conv1×1}$.

loss_function = CrossEntropyLoss(), optimizer = SGD, momentum = 0.9, weight_decay = 5e−4, lr_scheduler = MultiStepLR, milestones = [60, 110, 160], gamma = 0.1, stride = 1.

### Conv3×3 *vs* Conv1×1

In the following experiments, 16 variant structures were designed (Table 3), and the performance of variant structures decreased overall when replacing Conv3×3 with Conv1×1 as a shortcut. The settings are as layers = [17, 23], "β" = [1, 0], $width_{list}$ = {$w_1$, $w_2$, $w_3$, $w_4$}.

Some comparison between Conv3×3 and Conv1×1 is shown as follows. (1) Decrease in accuracy. Under the setting of the depth = 17 and 23, the max values of $Acc_{conv3×3}$ − $Acc_{conv1×1}$ were 0.46 and 0.34, the averages were 0.26 and 0.17% respectively. (2) The cut of params. The minimum of $Params_{conv3×3}$ − $Params_{conv1×1}$ is 2.25(M) and the max is 9.31 (M). The max cut ratio was 54% and the average was 34% (Fig. 3). (3) The cut ratio of FLOPs. The max cut ratio of FLOPs is 48%, and the average was 29% (Fig. 3). We can see from the comparison of experiments on the CIFAR-10 (*Krizhevsky, 2009*) dataset that the performance of Conv3×3 is better than that of Conv1×1 (Table 3). However, Conv1×1 has advantages in params and FLOPs.

In general, the number of parameters params and FLOPs will increase as the depth and width increase. Since the number of shortcuts is unchanging in TbsNet, the proportion of params and FLOPs decreasing will show a decreasing trend with increasing network depth

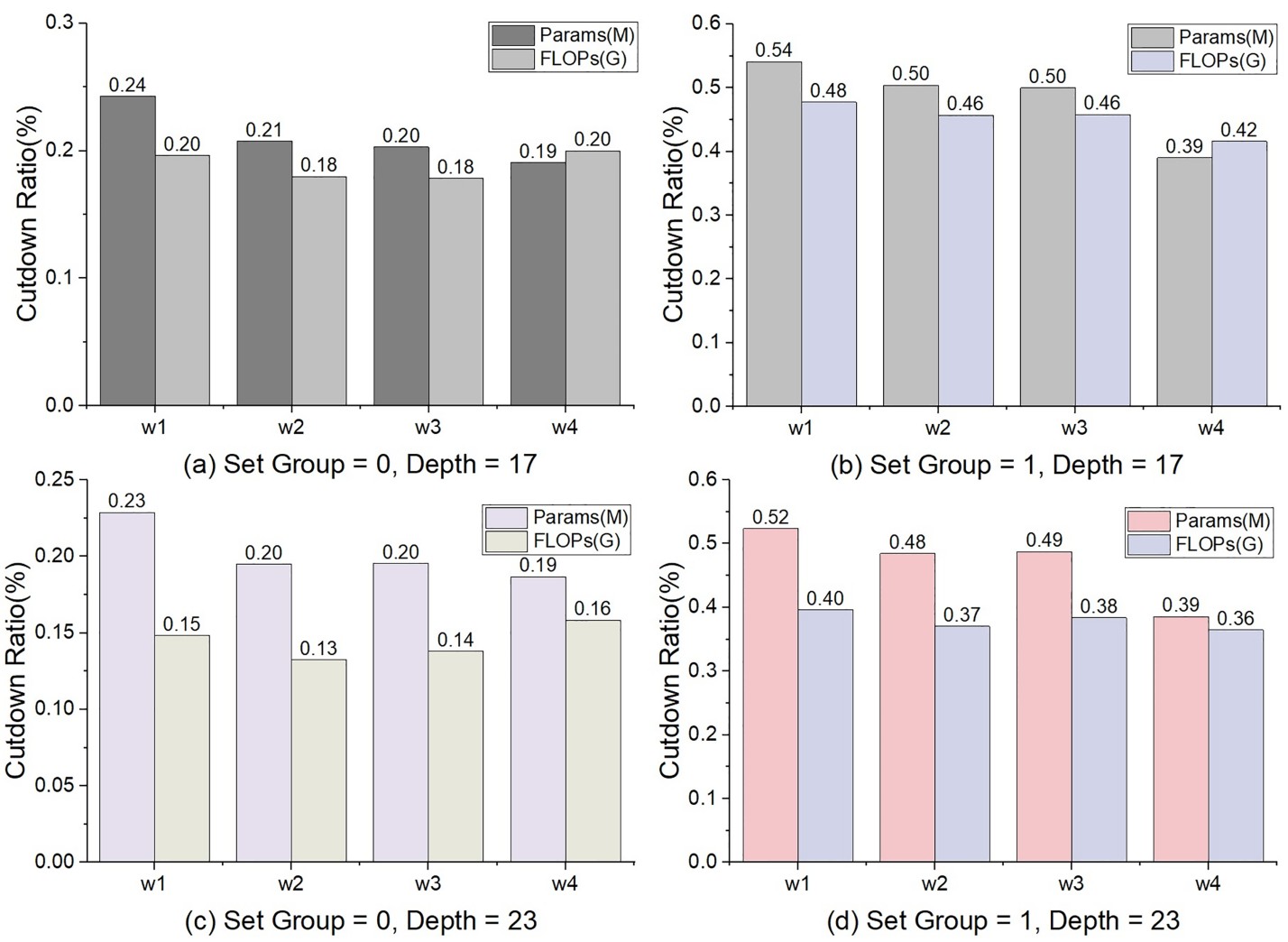

**Figure 3 The percentage of decline in params and FLOPs.** The calculation method is $\text{Params}_{\text{cutdown}} = (\text{Params}_{\text{Conv3}\times3} - \text{Params}_{\text{Conv1}\times1})/\text{Params}_{\text{Conv3}\times3}$, and so is to FLOPs.

and width (Fig. 3). When the asymmetric convolution in the stack unit sets the number of groups ("set_group" = 1) (Fig. 1), the total number of params and FLOPs of TbsNet will be drastically cut, and the parameters of the shortcut unit are still unchanged. Thus, the parameter proportion of the shortcut in the whole network is improved. That is why the cut ratio of params and FLOPs is significantly higher when the asymmetric convolution in stack unit sets the number of groups.

To sum up, two conclusions can be summarized. (1) Conv1×1 has the advantage of decreasing the parameters and floating-point computations, but Conv3×3 outperforms Conv1×1 in general. (2) It is beneficial to use Conv1×1 to reduce the number of parameters and floating-point computations. Therefore, TbsNet retains Conv1×1 as the lightweight option but prefers to use Conv3×3 as the short-hop connection.

## Asy *vs* GAsy

Replacing standard convolution with asymmetric convolution (*Ronneberger, Fischer & Brox, 2015*) can significantly reduce the number of parameters while maintaining stable performance (*Ding et al., 2019*). To further reduce the number of params and FLOPs, the groups of the first and second asymmetric convolution (Conv1×3 + Conv3×1) in stack unit (Fig. 1B) are assigned as 4 and 8, respectively. The settings of the network are as follows. shortcut = Conv3×3, Depth = $depth_{list}$, Width = $width_{list}$, β = [−1, 1], dataset = [CIFAR-10, CIFAR-100] (*Krizhevsky, 2009*). Selected expansion rate β = 1. Table 4 lists the Asy and GAsy results of experiments, where Asy denotes asymmetric convolution and GAsy denotes group asymmetric convolution.

We can see from the experimental data in Table 4 that compared with the Asy, the Top 1 Accuracy of the GAsy on CIFAR-10 and CIFAR-100 datasets (*Krizhevsky, 2009*) is reduced by 0.26 and 0.60 points on average, and the maximum reduction is 0.61 and 2.06 points. Of the 20 variants with different depths and widths, four variants improve their accuracy. It mainly appears in variant structures with larger network width, width = $w_3$ and $w_4$. Thus, it shows that adopting GAsy maybe decrease the performance compared to Asy overall. But in terms of parameters comparison, $Params_{GAsy}$ is much lower than $Params_{Asy}$, in which the maximum reduction ratio reaches 63%, and the average reduction rate is 58%. As param's decrease, the FLOPs decrease significantly.

Figure 4 shows the changing trend of $Acc_{GAsy}$ and $Acc_{Asy}$. In Fig. 4, the vertical axis represents the value of GAsy-Asy, and the horizontal axis is the sequence of 20 variant structures for the combination of $depth_{list}$ and $width_{list}$. Under the setting of β = −1, we observed that three values calculated by $Acc_{GAsy}$-$Acc_{Asy}$ are higher than zero, and all of those are experimental results on CIFAR-10 (*Krizhevsky, 2009*); on the other hand, all values ($Acc_{GAsy}$-$Acc_{Asy}$) on CIFAR-100 (*Krizhevsky, 2009*) are less than zero (for the experimental results of β = −1, please refer to Appendix 1). By comparing Figs. 4A and 4B, we found that the value change has a two-point similarity. One is that the values ($Acc_{GAsy}$ − $Acc_{Asy}$) are generally less than zero, and the other is that the values are gradually approaching zero with the increase of the depth and width of the network.

To sum up, two conclusions can be summarized. (1) GAsy will reduce the accuracy of the network, but with the increase of the depth and width of the network, the performance of GAsy will be gradually close to the Asy's. (2) Although GAsy can decrease the params and FLOPs significantly, it can not be adopted freely because of the depth and width of the network. When the depth and width of the network are small, it is not a good time to set the number of groups for asymmetric convolution, but the opportunity is when the depth or width of the network is high.

According to the above conclusions, the TBS block retains the setting of the number of groups as an option. It is worth mentioning that group convolution and depthwise separable convolution may be affected by cuDNN and lead to increase training and inference time.

**Table 4 Comparison of three performance indicators (params, FLOPs, and accuracy) between GAsy and Asy with expansion coefficient β = 1.**

| Depth (Layers) | Width | Params (M) | | | FLOPs (G) | | | Top 1 Acc (%) | | | | | |
| --- | --- | --- | --- | --- | --- | --- | --- | --- | --- | --- | --- | --- | --- |
| | | | | | | | | CIFAR-10 | | | CIFAR-100 | | |
| | | Asy | GAsy | (%↓) | Asy | GAsy | (%↓) | Asy | GAsy | (↑↓) | Asy | GAsy | (↑↓) |
| 17 | $w_1$ | 9.27 | 4.16 | 0.55 | 35.12 | 14.43 | 0.59 | 95.92 | 95.66 | −0.26 | 81.03 | 79.25 | −1.78 |
| | $w_2$ | 15.95 | 6.59 | 0.59 | 56.49 | 22.22 | 0.61 | 96.48 | 95.91 | −0.57 | 81.72 | 80.22 | −1.5 |
| | $w_3$ | 26.18 | 10.63 | 0.59 | 68.66 | 26.74 | 0.61 | 96.38 | 96.14 | −0.24 | 81.91 | 80.86 | −1.05 |
| | $w_4$ | 48.82 | 23.87 | 0.51 | 76.70 | 36.82 | 0.52 | 96.11 | 96.01 | −0.10 | 81.35 | 81.71 | 0.36† |
| 23 | $w_1$ | 9.84 | 4.3 | 0.56 | 46.48 | 17.39 | 0.63 | 96.31 | 96.01 | −0.30 | 81.52 | 80.31 | −1.21 |
| | $w_2$ | 16.97 | 6.85 | 0.60 | 76.61 | 27.41 | 0.64 | 96.75 | 96.21 | −0.54 | 82.28 | 81.23 | −1.05 |
| | $w_3$ | 27.21 | 10.89 | 0.60 | 88.78 | 31.94 | 0.64 | 96.68 | 96.1 | −0.58 | 82.08 | 82.07 | −0.01 |
| | $w_4$ | 49.84 | 24.13 | 0.52 | 96.82 | 42.01 | 0.57 | 96.46 | 96.31 | −0.15 | 81.55 | 81.71 | $0.16^{\dagger}$ |
| 27 | $w_1$ | 10.42 | 4.45 | 0.57 | 57.84 | 20.35 | 0.65 | 96.73 | 96.12 | −0.61 | 82.43 | 80.37 | −2.06 |
| | $w_2$ | 17.99 | 7.11 | 0.60 | 96.73 | 32.60 | 0.66 | 96.66 | 96.39 | −0.27 | 82.67 | 81.9 | −0.77 |
| | $w_3$ | 28.23 | 11.15 | 0.61 | 108.91 | 37.13 | 0.66 | 96.79 | 96.55 | −0.24 | 82.92 | 82.7 | −0.22 |
| | $w_4$ | 50.86 | 24.39 | 0.52 | 116.94 | 47.20 | 0.60 | 96.64 | 96.47 | −0.17 | 82.72 | 82.27 | −0.45 |
| 35 | $w_1$ | 11.34 | 4.68 | 0.59 | 69.14 | 23.26 | 0.66 | 96.81 | 96.34 | −0.47 | 82.26 | 81.6 | −0.66 |
| | $w_2$ | 19.62 | 7.52 | 0.62 | 116.78 | 37.72 | 0.68 | 96.77 | 96.65 | −0.12 | 83.31 | 82.49 | −0.82 |
| | $w_3$ | 29.86 | 11.57 | 0.61 | 128.96 | 42.25 | 0.67 | 96.86 | 96.73 | −0.13 | 83.29 | 82.59 | −0.7 |
| | $w_4$ | 52.49 | 24.81 | 0.53 | 164.58 | 52.32 | 0.68 | 96.71 | 96.88 | $0.17^{\dagger}$ | 83.53 | 82.88 | −0.65 |
| 44 | $w_1$ | 12.72 | 5.04 | 0.60 | 86.10 | 27.62 | 0.68 | 96.69 | 96.4 | −0.29 | 82.56 | 82.29 | −0.27 |
| | $w_2$ | 22.07 | 8.14 | 0.63 | 146.86 | 45.40 | 0.69 | 97.02 | 96.67 | −0.35 | 82.99 | 82.78 | −0.21 |
| | $w_3$ | 32.31 | 12.19 | 0.62 | 159.03 | 49.93 | 0.69 | 96.81 | 96.93 | $0.12^{\dagger}$ | 82.51 | 83.53 | $1.02^{\dagger}$ |
| | $w_4$ | 54.94 | 25.43 | 0.54 | 167.06 | 60.00 | 0.64 | 96.71 | 96.62 | −0.09 | 83.32 | 83.24 | −0.08 |
| Max | | 54.94 | 25.43 | 0.63 | 167.06 | 60.00 | 0.69 | 97.02 | 96.93 | 0.17 | 83.53 | 83.53 | 1.02 |
| Min | | 9.27 | 4.16 | 0.51 | 35.12 | 14.43 | 0.52 | 95.92 | 95.66 | −0.61 | 81.03 | 79.25 | −2.06 |
| Average | | 27.35 | 11.90 | 0.58 | 98.23 | 34.74 | 0.64 | 96.61 | 96.36 | −0.26 | 82.40 | 81.80 | −0.60 |
| Std | | 15.41 | 7.70 | 0.04 | 38.65 | 12.18 | 0.04 | 0.26 | 0.33 | 0.21 | 0.70 | 1.10 | 0.72 |

Notes:

Accuracy (↑↓) = $Acc_{GAsy}$ − $Acc_{Asy}$.

† Denotes $Acc_{GAsy}$ is greater than $Acc_{Asy}$. (%↓) Denotes the cut ratio of params and FLOPs. (↑↓) Denotes that if the value is positive means higher else lower.

## Scaling ability

The verification of the scaling ability lies in four aspects. (1) Network depth. The TBS blocks of each Stage are stacked to expand the depth of the network. (2) The width of the network. The widths of every stage are set according to the width$_{list}$. (3) Expansion and squeeze mechanism. The expansion coefficient β in the TBS block completes the scaling of the network width (4) Feature map scaling. The downsampling multiple is controlled by the stride, with a minimum of 8X and a maximum of 32X to adapt to feature maps of different sizes. The feature map size plays a crucial role in determining the scaling ability of the network. If the feature maps are too small, it would severely limit the network's ability to scale and tackle large-sized input images. In contrast, while larger feature maps can offer richer information, they can also make the network excessively complex and bloated, resulting in delayed training and prediction speeds, which may not be practical. In this

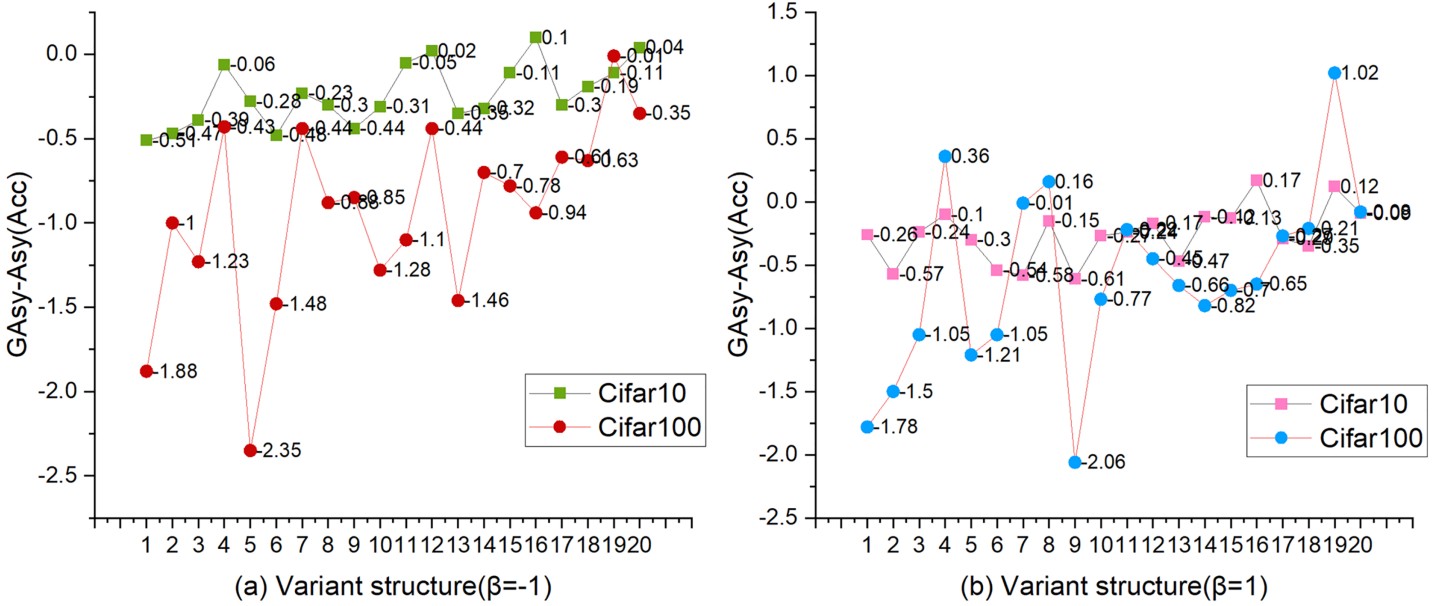

**Figure 4 The comparison between Acc_Asy and Acc_GAsy.** B = [−1, 1], dataset = [CIFAR-10, CIFAR-100], shortcut = Conv3×3. The vertical axis is the value of $Acc_{GAsy} - Acc_{Asy}$, and the horizontal axis is the sequence of 20 variant structures with different depths and widths.

experiment, the params, FLOPs, and accuracy (%) still indicate the scaling ability of TbsNet (*Dollár, Singh & Girshick, 2021*).

We selected a base model as the baseline, and the settings are as follows. Depth = 17, width = $w_1$, shortcut = Conv3×3, set_group = 1, β = −1. Depth_list = {17, 23, 29, 35, 44} and width_list = {$w_1$, $w_2$, $w_3$, $w_4$} are the depth and width lists for the scaling experiment. The number of parameters of the baseline is params = 3.65M, the number of floating-point calculations is PLOPs = 12.42G, the Top 1 Accuracy on CIFAR-10 (*Krizhevsky, 2009*) is $Acc_{CIFAR-10}$ = 95.49, and on CIFAR-100 (*Krizhevsky, 2009*) is $Acc_{CIFAR-100}$ = 78.31.

### Scale the depth

When depth = 44 and width = $w_1$, three performance indicators get significant changes, the params = 4.53M, the PLOPs = 25.61G, the Top 1 Accuracy on CIFAR-10 (*Krizhevsky, 2009*) $Acc_{CIFAR-10}$ = 96.36 and on CIFAR-100 (*Krizhevsky, 2009*) $Acc_{CIFAR-100}$ = 81.43 (Table 5). Compared with the baseline, the params is not increased significantly with the increase of depth, the FLOPs increased relatively more, and the accuracy is also greatly improved.

### Expand the width

When depth = 17, width = $w_4$, three performance indicators also get significant changes, the params = 16.62M, the PLOPs = 28.14G, the Top 1 Accuracy on CIFAR-10 (*Krizhevsky, 2009*) $Acc_{CIFAR-10}$ = 95.92 and on CIFAR-100 (*Krizhevsky, 2009*) $Acc_{CIFAR-100}$ = 80.81 (Table 5). Compared with the baseline model, the params and FLOPs increased highly, and the accuracy improved greatly.

**Table 5** The influence of the depth and width of the network on the three performance indicators of params, FLOPs, and accuracy.

| Properties | Depth (Layers) | Width | | | | Max | Min | Average | Std |
|---|---|---|---|---|---|---|---|---|---|
| | | $w_1$ | $w_2$ | $w_3$ | $w_4$ | | | | |
| Params (M) | 17 | 3.65 | 5.46 | 8.64 | 16.62 | 16.62 | 3.65 | 8.59 | 4.97 |
| | 23 | 3.79 | 5.72 | 8.90 | 16.88 | 16.88 | 3.79 | 8.82 | 5.00 |
| | 29 | 3.94 | 5.98 | 9.16 | 17.14 | 17.14 | 3.94 | 9.06 | 5.02 |
| | 35 | 4.17 | 6.39 | 9.57 | 17.56 | 17.56 | 4.17 | 9.42 | 5.08 |
| | 44 | 4.53 | 7.01 | 10.19 | 18.18 | 18.18 | 4.53 | 9.98 | 5.14 |
| FLOPs (G) | 17 | 12.42 | 18.72 | 22.09 | 28.14 | 28.14 | 12.42 | 20.35 | 5.68 |
| | 23 | 15.39 | 23.91 | 27.29 | 33.33 | 33.33 | 15.39 | 24.98 | 6.49 |
| | 29 | 18.35 | 29.10 | 32.48 | 38.53 | 38.53 | 18.35 | 29.61 | 7.33 |
| | 35 | 21.25 | 34.22 | 37.60 | 43.65 | 43.65 | 21.25 | 34.18 | 8.19 |
| | 44 | 25.61 | 41.90 | 45.28 | 51.33 | 51.33 | 25.61 | 41.03 | 9.52 |
| Acc (%) CIFAR-10 | 17 | 95.49 | 95.81 | 95.91 | 95.92 | 95.92 | 95.49 | 95.78 | 0.17 |
| | 23 | 95.91 | 96.07 | 96.37 | 96.16 | $96.37^{\ddagger}$ | 95.91 | 96.13 | 0.17 |
| | 29 | 96.14 | 96.27 | 96.55 | 96.40 | $96.55^{\ddagger}$ | 96.14 | 96.34 | 0.15 |
| | 35 | 96.29 | 96.43 | 96.59 | 96.81 | 96.81 | 96.29 | 96.53 | 0.19 |
| | 44 | 96.34 | 96.62 | 96.83 | $96.61\downarrow$ | $96.83^{\ddagger}$ | 96.34 | 96.60 | 0.17 |
| Acc (%) CIFAR-100 | 17 | 78.31 | 79.82 | 79.95 | 80.81 | 80.81 | 78.31 | 79.72 | 0.90 |
| | 23 | 79.22 | 80.69 | 81.44 | 81.06 | $81.44^{\ddagger}$ | 79.22 | 80.60 | 0.84 |
| | 29 | 80.86 | 81.35 | 81.80 | 82.36 | 82.36 | 80.86 | 81.59 | 0.55 |
| | 35 | 81.05 | 82.16 | 82.53 | 82.62 | 82.62 | 81.05 | 82.09 | 0.62 |
| | 44 | 81.43 | 82.41 | 82.93 | 82.86 | $82.93^{\ddagger}$ | 81.43 | 82.41 | 0.60 |

**Note:**
Max, min, average, and standard deviation (Std) are the maximum value, minimum value, average value, and standard deviation of the three indicators of params, FLOPs, and accuracy, respectively. '$\ddagger$' and '$\downarrow$' respectively indicate that the current $Acc_{width}$ or $Acc_{depth}$ is lower than its previous one when the depth or width remains unchanged.

### Depth and width

We can see from Table 5 that the increase of params by expanding the network width is much higher than by increasing the network depth. But the two approaches are very close in terms of FLOPs. The improvement of accuracy by expanding the width is not as good as increasing the depth. The three performance indicators (params, FLOPs, and accuracy) will get signally variations when the depth and width are enlarged together. The variations are close to or slightly lower than the sum of network depth and width alone.

### The maximum of width and depth

Increasing the depth and width of the network at the same time can further improve accuracy, but the increment in terms of params and FLOPs is signal. We find that expanding the width brings instability to the improvement of accuracy. There are four accuracy drops in Table 5 due to the width expansion.

Figures 5B–5F more intuitively shows the impact of expanding width on the three performance indicators of params, FLOPs, and accuracy. Under the setting of width = $w_4$, there were six accuracy drops and four raises. Four of the six accuracy drops occurred in

experiments of CIFAR-10 datasets (*Krizhevsky, 2009*). A similar phenomenon occurs in the experiment with β = 1 (Table 4). Experiments show that the setting of width = $w_4$ cannot improve the performance of TbsNet on the CIFAR-10 and CIFAR-100 datasets (*Krizhevsky, 2009*), especially on the CIFAR-10 datasets (*Krizhevsky, 2009*).

In terms of network depth selection, Fig. 5A shows the effect of depth on network performance. As the depth increases, the rate of accuracy improvement gradually stabilizes while FLOPs still increase significantly, so layers = 44 is retained as the maximum depth, and no deeper network is constructed.

### Expansion and squeeze

ES is a more concise way of channel information fusion than channel attention mechanism. In our design scheme, the dimension size of the feature information is controlled by the selection of the expansion coefficient, so as to control the change of the performance three performance indicators. It is also a means to test the ability of network scaling and network lightweight. The expansion coefficient β also affects the performance of the network. Respectively, Tables 4, 5, Figs. 4, and 5 shows the statistics of the three performance indicators of params, FLOPs, and accuracy under the setting of β = 1 and β = −1, and the overall TbsNet β = 1 the performance with β = 1 is significantly better than β = −1. So TbsNet is more inclined to choose β = 1 and retains the β = −1 option as a lightweight method.

## BENCHMARKING

We will verify the performance difference between TbsNet and various architectures in experiments. The main benchmarking platforms are CIFAR-10, CIFAR-100 (*Krizhevsky, 2009*), TinyImageNet (*Le & Yang, 2015*), ImageNet-1K (*Russakovsky et al., 2015*), *etc*. The evaluation indicators are depth, width, params, FLOPs, and accuracy (*Zhang et al., 2023*). Some previous state-of-the-art network architectures are selected for comparison, such as ResNets (*He et al., 2016b*), RepVGG (*Ding et al., 2021*), DenseNet (*Huang et al., 2017*), ResNexts (*Xie et al., 2017*), WideResNet (*Zagoruyko & Komodakis, 2016*), SENet (*Hu et al., 2020*), ParNet (*Goyal et al., 2021*), MobileV2 (*Sandler et al., 2018*), EfficientNet (*Tan & Le, 2019*), EfficientNetV2 (*Tan & Le, 2021*), *etc*. The basic experimental environment is Python3.8.6, Pytorch1.8.2, Cuda11.2, and NVIDIA 3090 GPU×3. The hyperparameter settings for training on the CIFAR-10 and CIFAR-100 datasets (*Krizhevsky, 2009*) are the same as in Section 4.

### On CIFAR datasets

The CIFAR-10 dataset (*Krizhevsky, 2009*) consists of 60,000 32×32 color images in 10 classes, with 50,000 training images and 10,000 test images. Each image is labeled as one of 10 classes, and each class consists of 5,000 training images and 1,000 test images. The CIFAR-100 dataset (*Krizhevsky, 2009*) consists of 60,000 32×32 color images and has 100 classes.

Table 6 summarizes the performance of various networks on CIFAR-10 and CIFAR-100. On the CIFAR-10 dataset, Top 1 Accuracy exceeds ResNet50 (*He et al., 2016b*), ResNet152 (*He et al., 2016b*), ResNeXt101 (32×8d) (*Xie et al., 2017*), WideResNet (28, 10)

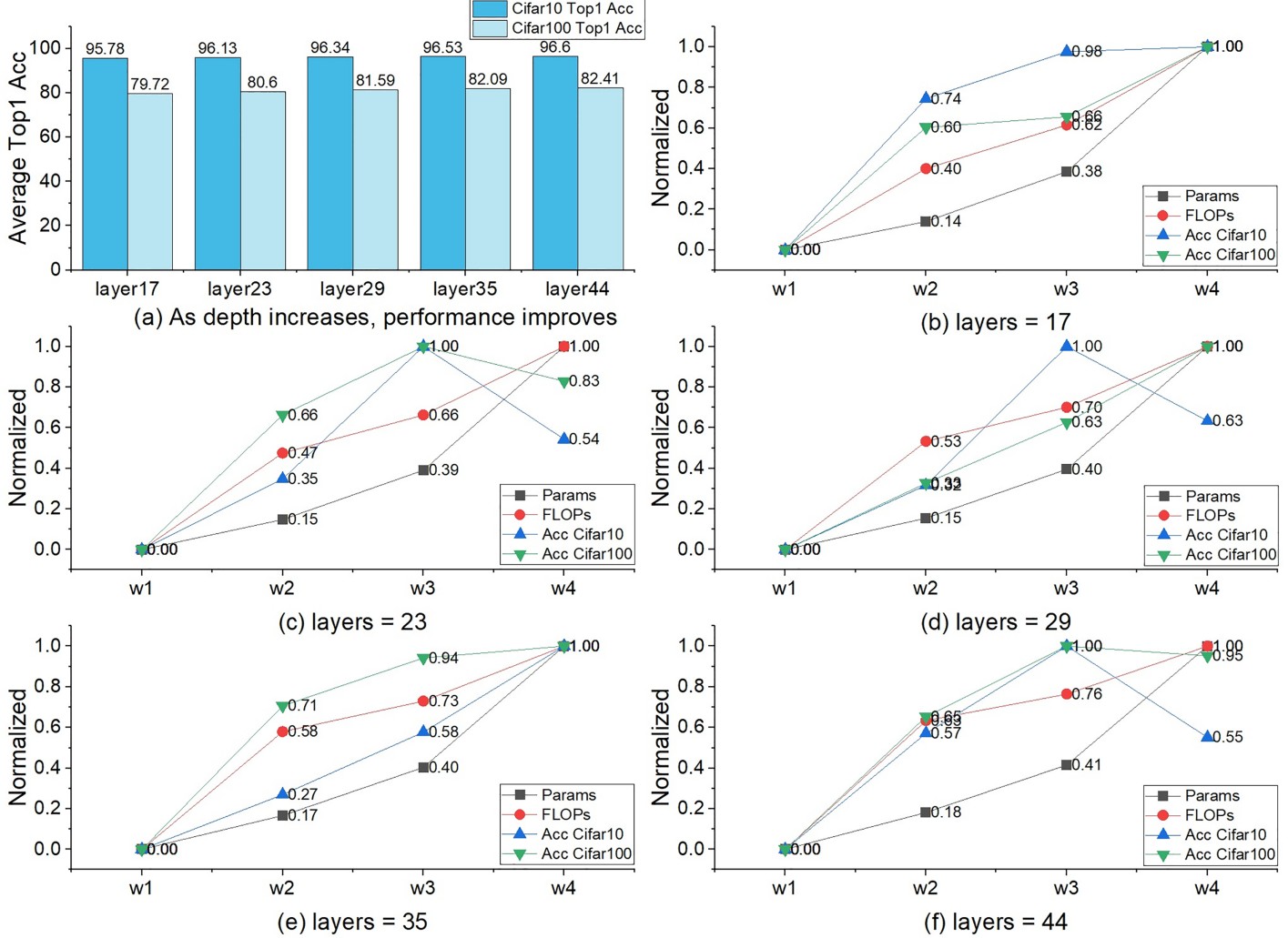

**Figure 5 The influence of network width on params, FLOPs, and accuracy.** The vertical axis represents the value after normalization (the calculation method is (X-Min)/(Max-Min)), and the horizontal axis represents the width$_{list}$, where X is the value of the three performance indicators params, FLOPs, and accuracy under different widths.

(Zagoruyko & Komodakis, 2016), RepVGG-B2g4 (Ding et al., 2021), Parnet (Goyal et al., 2021) by up to 2.48 points, 1.67 points, 1.24 points, 1.02 points, 0.7 points, 0.9 points, respectively. On the CIFAR-100 dataset, TbsNet's Top 1 Accuracy exceeds ResNet50 (He et al., 2016b), ResNet152 (He et al., 2016b), ResNeXt101 (32×8d) (Xie et al., 2017), WideResNet (28, 10) (Zagoruyko & Komodakis, 2016), RepVGG-B2g4 (Ding et al., 2021), Parnet (Goyal et al., 2021) by up to 7.97 points, 7.67 points, 2.72 points, 2.81 points, 1.84 points, 2.21 points, respectively.

From the above analysis, we sum up that TbsNet has the advantage of lower params and FLOPs, the performance of TbsNet is competitive with the most advanced CNNs, such as ResNet (He et al., 2016b), ResNeXt (Xie et al., 2017), and RepVgg (Ding et al., 2021), etc.

**Table 6 Comparison of performance indicators between TbsNet and other network architectures on the CIFAR dataset.**

| Architecture | Attention/β (Ours) | Depth (Layers) | Width range | Params (M) | FLOPs (G) | Top1 Acc (%) | |
|---|---|---|---|---|---|---|---|
| | | | | | | CIFAR-10 | CIFAR-100 |
| ResNet18 (*He et al., 2016a*) | × | 18 | [64, 512] | 11.15 | 1.7 | 93.41 | 75.61 |
| ResNet34 (*He et al., 2016b*) | × | 34 | [64, 512] | 20.75 | 3.43 | 94.22 | 76.76 |
| ResNet50 (*He et al., 2016b*) | × | 50 | [64, 2,048] | 24.37 | 3.86 | 94.39 | 77.39 |
| ResNet152 (*He et al., 2016b*) | × | 101 | [64, 2,048] | 57.41 | 10.83 | 95.35 | 77.69 |
| ResNeXt50 (32×4d) (*Xie et al., 2017*) | √ | 50 | [64, 2,048] | 23.87 | 4.01 | 94.66 | 77.77 |
| ResNeXt101 (32×8d) (*Xie et al., 2017*) | √ | 101 | [64, 2,048] | 84.67 | 15.43 | 95.78 | 80.83 |
| WideResNet (28, 10) (*Zagoruyko & Komodakis, 2016*) | × | 28 | [160, 640] | 36.5M | 239.81 | 96 | 80.75 |
| WideResNet (40, 4) (*Zagoruyko & Komodakis, 2016*) | × | 40 | [64, 256] | 8.56 | 59.58 | 95.47 | 78.82 |
| SEResNet50 (*Hu et al., 2020*) | √ | 50 | [64,2048] | 25.05 | 60.09 | 94.31 | 77.93 |
| SEResNet 152 (*Hu et al., 2020*) | √ | 152 | [64, 2,048] | 62 | / | 96.02 | 79.34 |
| RepVGG-B2g4 (*Ding et al., 2021*) | √ | 28 | [64, 2,560] | 55.77 | 113 | 96.32 | 81.72 |
| ParNet (*Goyal et al., 2021*) | √ | 12 | / | 15.5 | / | 96.01 | 79.98 |
| | √ | 12 | / | 35. | / | 96.12 | 81.35 |
| TbsNet (17, $w_1$) (Ours) | 1 | 17 | [64, 512] | 4.16 | 14.43 | 95.66 | 79.25 |
| TbsNet (17, $w_2$) (Ours) | −1 | 17 | [64, 768] | 5.46 | 18.72 | 95.81 | 79.82 |
| TbsNet (23, $w_1$) (Ours) | 1 | 23 | [64, 512] | 4.3 | 17.39 | 96.01 | 80.31 |
| TbsNet (23, $w_2$) (Ours) | −1 | 23 | [64, 768] | 5.72 | 23.91 | 96.07 | 80.69 |
| TbsNet (29, $w_2$) (Ours) | 1 | 29 | [64, 768] | 7.11 | 32.60 | 96.39 | 81.9 |
| TbsNet (29, $w_3$) (Ours) | −1 | 29 | [64, 1,024] | 9.16 | 32.48 | 96.55 | 81.8 |
| TbsNet (35, $w_2$) (Ours) | 1 | 35 | [64, 768] | 7.52 | 37.72 | 96.65 | 82.49 |
| TbsNet (35, $w_3$) (Ours) | −1 | 35 | [64, 1,024] | 9.57 | 37.60 | 96.59 | 82.53 |
| TbsNet (44, $w_2$) (Ours) | 1 | 44 | [64, 1,024] | 8.14 | 45.40 | 96.67 | 82.78 |
| TbsNet (44, $w_3$) (Ours) | −1 | 44 | [64, 1,024] | 10.19 | 45.28 | 96.83 | 82.93 |
| TbsNet (CIFAR-10$_{best}$) (Ours) | 1 | 44 | [64, 1,024] | 22.07 | 146.86 | 97.02 | / |
| TbsNet (CIFAR-100$_{best}$) (Ours) | 1 | 35 | [64, 2,048] | 28.51 | 109.39 | / | 83.56 |

**Note:**
Params and FLOPs of all architectures are recalculated by torchsummary and fvcore. The settings of TbsNet are as follows. Set_group = 1, shortcut = Conv3×3. '/' is marked as the original article did not describe it clearly. TbsNet (17, $w_1$) in the table refers to a variant structure with a depth of 17 and a width of $w_1$.

## On Tiny-ImageNet dataset

Tiny-ImageNet (*Le & Yang, 2015*) has 200 classes, each class has 500 training images, 50 validation images, and 50 test images, and the size of the images is 64×64. The experimental environment is the same as the hyperparameter setting of the CIFAR dataset and training with the Vanilla (*Abai & Rajmalwar, 2019*) method.

We compared some previous state-of-the-art network architectures with TbsNet (Table 7). These architectures employ some training techniques such as AutoMix (*Liu et al., 2022c*), SAMix (*Li et al., 2021*), PuzzleMix+DM (*Liu et al., 2022b*), DCL (*Luo et al., 2021*), *etc*. Since no training techniques are employed (vanilla training scheme), the TbsNet

**Table 7 Comparison with previous state-of-the-art architectures on the Tiny-ImageNet dataset.**

| Architecture | Tricks & Methods | Top1 Acc (%) |
|---|---|---|
| DenseNet + Residual Networks | Vanilla (*Abai & Rajmalwar, 2019*) | 60 |
| ResNet18 | AutoMix (*Liu et al., 2022c*) | 67.33 |
| | SAMix (*Li et al., 2021*) | 68.89 |
| ResNeXt-50 | AutoMix (*Liu et al., 2022c*) | 70.72 |
| | PuzzleMix+DM (*Liu et al., 2022b*) | 68.04 |
| | SAMix (*Li et al., 2021*) | 72.18 |
| EfficientNet-B1 | DCL (*Luo et al., 2021*) | 84.39 |
| TbsNet (17, $w_2$) (Ours) | Vanilla (*Abai & Rajmalwar, 2019*) | 67.7 |
| TbsNet (23, $w_2$) (Ours) | Vanilla (*Abai & Rajmalwar, 2019*) | 68.95 |
| TbsNet (23, $w_3$) (Ours) | Vanilla (*Abai & Rajmalwar, 2019*) | 69.57 |
| TbsNet (29, $w_3$) (Ours) | Vanilla (*Abai & Rajmalwar, 2019*) | 69.88 |
| TbsNet (35, $w_3$) (Ours) | Vanilla (*Abai & Rajmalwar, 2019*) | 70.27 |
| TbsNet (44, $w_3$) (Ours) | Vanilla (*Abai & Rajmalwar, 2019*) | 70.89 |

**Note:**

TbsNet (17, $w_2$) refers to a variant structure with a depth of 17 and a width of $w_2$.

results on Tiny-ImageNet (*Le & Yang, 2015*) are not very nice. Compared with these architectures TbsNet still shows excellent performance. The settings of the network model are $\beta = -1$, set_group = 0, and shortcut = Conv3×3.

## On ImageNet-1K dataset

The ImageNet-1K dataset (*Russakovsky et al., 2015*) contains 14,197,122 labeled images. Since 2010, this dataset has been used in the ImageNet Large-Scale Visual Recognition Challenge (ILSVRC), a benchmark for image classification and object detection (*Russakovsky et al., 2015*). The ImageNet-1K dataset is a sub-dataset used in the ISLVRC2012 competition (*Hariharan & Girshick, 2017*). The ImageNet-1K dataset (*Russakovsky et al., 2015*) has a total of 1,000 categories. The training set has 1,281,167 pictures, about 1,300 pictures in each category; the validation set has 50,000 pictures, 50 pictures in each category, and the test set has 100,000 pictures, 100 pictures in each category.

Employing different training schemes will yield different accuracies on the ImageNet-1K dataset (*Wightman, Touvron & Jégou, 2021*). Table 8 lists the weights obtained from Pytorch official ResNet50, MobileNetV2, and other network architectures under two different training schemes, namely IMAGENET1K_V1 and IMAGENET1K_V2 (*Torch Contributors, 2017*). The test accuracy of IMAGENET1K_V2 is higher than IMAGENET1K_V1. The accuracy obtained under the training scheme of *Goyal et al. (2021)* are between the IMAGENET1K_V1 and IMAGENET1K_V2. The training schemes are given by *Liu et al. (2022a)*, *Foret et al. (2020)* are significantly better than the test accuracy of IMAGENET1K_V1 and even better than IMAGENET1K_V2's.

We employed distributed multi-GPU and mixed-precision training schemes to train the model. Data enhancement techniques and learning rate adjustment strategies were selected (*Wightman, Touvron & Jégou, 2021*). The training program we adopted refers

**Table 8 Accuracy comparison on the ImageNet-1K dataset.**

| Architecture | Weights, main tricks and methods | Top 1 Acc (%) |
|---|---|---|
| ResNet50[†] | IMAGENET1K_V1 (*Torch Contributors, 2017*) | 76.13 |
| | IMAGENET1K_V2 (*Torch Contributors, 2017*) | 80.86 |
| ResNet50 (*Goyal et al., 2021*) | Cropping, flipping, color-jitter, and rand-augment | 77.53 |
| ResNet152[†] | IMAGENET1K_V1 (*Torch Contributors, 2017*) | 78.31 |
| | IMAGENET1K_V2 (*Torch Contributors, 2017*) | 82.28 |
| ResNeXt101_32×8[†] | IMAGENET1K_V1 (*Torch Contributors, 2017*) | 79.31 |
| | IMAGENET1K_V2 (*Torch Contributors, 2017*) | 82.83 |
| ParNet-S (*Goyal et al., 2021*) | Cropping, flipping, color-jitter, and rand-augment | 75.19 |
| ParNet-XL (*Goyal et al., 2021*) | Cropping, flipping, color-jitter, and rand-augment | 78.55 |
| | +Longer Training,Train & Test Res. 320,10-crop testing) | 80.72 |
| MobileNet_V2[†] | IMAGENET1K_V1 (*Torch Contributors, 2017*) | 71.88 |
| MobileNet_V2[†] | IMAGENET1K_V2 (*Torch Contributors, 2017*) | 72.15 |
| ConvNeXt_Tiny[†] | IMAGENET1K_V1 (*Torch Contributors, 2017*) | 82.52 |
| ConvNeXt_Base[†] | IMAGENET1K_V1 (*Torch Contributors, 2017*) | 84.06 |
| ConvNeXt_Large[†] | IMAGENET1K_V1 (*Torch Contributors, 2017*) | 84.41 |
| ConvNeXt-XL | AdamW, warmup, Mixup, Cutmix, Label Smoothing, et al. (*Liu et al., 2022a*) | 87.8 |
| EfficientNet_B0[†] | IMAGENET1K_V1 (*Torch Contributors, 2017*) | 77.69 |
| EfficientNet_B7[†] | IMAGENET1K_V1 (*Torch Contributors, 2017*) | 84.12 |
| EfficientNet_V2_S[†] | IMAGENET1K_V1 (*Torch Contributors, 2017*) | 84.23 |
| EfficientNet_V2_M[†] | IMAGENET1K_V1 (*Torch Contributors, 2017*) | 85.11 |
| EfficientNet_V2_L[†] | IMAGENET1K_V1 (*Torch Contributors, 2017*) | 85.81 |
| EfficientNet-L2-475 (*Foret et al., 2020*) | SAM to finetuning EfficentNet (pretrained on ImageNet) | 88.61 |
| TbsNet (17, $w_2$) (Ours) | Long Training, Random Erasing, Cutmix, EMA, et al. (*Vryniotis, 2021*) | 83.13 |
| TbsNet (23, $w_2$) (Ours) | Long Training, Random Erasing, Cutmix, EMA, et al. (*Vryniotis, 2021*) | 83.89 |
| TbsNet (23, $w_3$) (Ours) | Long Training, Random Erasing, Cutmix, EMA, et al. (*Vryniotis, 2021*) | 84.41 |
| TbsNet (29, $w_3$) (Ours) | Long Training, Random Erasing, Cutmix, EMA, et al. (*Vryniotis, 2021*) | 85.16 |
| TbsNet (35, $w_3$) (Ours) | Long Training, Random Erasing, Cutmix, EMA, et al. (*Vryniotis, 2021*) | 85.59 |
| TbsNet (44, $w_3$) (Ours) | Long Training, Random Erasing, Cutmix, EMA, et al. (*Vryniotis, 2021*) | 86.17 |

**Notes:**
TbsNet (17, $w_2$) refers to a variant structure with a depth of 17 and a width of $w_2$.
[†] Annotated accuracies reported on ImageNet-1K in Torchvision.

to the setting in Pytorch's official blog "How to Train State-Of-The-Art Models Using TorchVision's Latest Primitives" (*Vryniotis, 2021*), which is equivalent to IMAGENET1K_V2, so it is a relatively fair comparison to IMAGENET1K_V2.

The experimental data in Table 8 shows that TbsNet exceeds ResNet50_IMAGENET1K_V2, ResNet152_IMAGENET1K_V2, ResNeXt101_32×8D_IMAGENET1K_V2, by up to 5.31 points, 3.89 points, 3.34 points, respectively. Compared with the ParNet (*Goyal et al., 2021*) and ConvNeXt (*Liu et al., 2022a*), TbsNet exceeds ParNet by about four points and exceeds ConvNeXt_Large_IMAGENET1 by up to 1.3 points. Although The Top 1 Accuracy of TbsNet is equivalent to ConvNeXts (*Liu et al., 2022a*) and EfficientNet_V2_ IMAGENET1K_V1, TbsNet has more advantages in terms of params and FLOPs.

## MEDICAL IMAGE SEGMENTATION

We design a simple U-shaped network model Tbs-UNet with TbsNet as the backbone network and complete the medical image segmentation task on the Synapse dataset (*Chen et al., 2021*). The encoder and decoder of the U-shaped network architecture adopt four stages, respectively. Channels change list is {$C_1 = 32$, $C_2 = 64$, $C_3 = 128$, $C_4 = 256$, $C_5 = 512$}. The encoder consists of four cascaded stages, and each stage realizes the 2X downsampling. The stack number list of the stages is [1, 2, 3, 1], with twenty-twolayers. In the middle of the Tbs-UNet structure, there is a bottleneck consisting of two full connection layers. The decoder consists of four cascaded stages corresponding to the encoder, and each stage realizes the 2X upsampling. There are four skip connections between the upsampling and the downsampling. The input feature map size is 224×224.

We use 30 abdominal CT scans from the MICCAI 2015 Multi-Atlas Abdomen Labeling Challenge (https://www.synapse.org/#!Synapse:syn3193805/wiki/218292). A total of 3,779 axial contrast-enhanced abdominal clinical CT images were obtained (*Chen et al., 2021*). Each CT volume consisted of 85–198 slices of $512 \times 512$ pixels with a spatial resolution of ([0.54–0.54] [0.98–0.98] [2.5–5.0]) mm$^3$ voxel. TbsNet achieved the experimental results with an average dice score (DSC) of 78.39% and an average Hausdorff distance (HD) of 31.13 (mm). With the same input feature map size, Tbs-UNet's DSC is higher than TransUNet's 0.91 percentage points, and HD is lower than 0.56 (mm) (*Chen et al., 2021*). In terms of inference time, Tbs-UNet is equivalent to TransUNet, but memory usage is about 7.6 percentage points higher than TransUNet (*Chen et al., 2021*).

Through the medical image segmentation experiments implemented on the Synapse dataset, we prove that TbsNet is also suitable for some downstream tasks of computer vision.

## CONCLUSIONS

We analyzed several notable factors affecting network performance and proposed some strategies for constructing CNNs. An efficient convolutional neural network architecture should achieve the optimal combination of factors such as network depth, width, receptive field, parameter amount, and inference time. To this end, we propose strategies for building an efficient convolutional neural network and designing TbsNet with a thin-branch structure. We have done extensive experiments on benchmarks such as CIFAR-10 (*Krizhevsky, 2009*), CIFAR-100 (*Krizhevsky, 2009*), Tiny-ImageNet (*Le & Yang, 2015*), and ImageNet-1K (*Russakovsky et al., 2015*) to verify the performance of TbsNet. Compared with the previous state-of-the-art architectures, TbsNet has the advantages of a simple structure, less branch structure, and fast inferring. TbsNet does not need reparameterization, and it can be competent for some downstream tasks in computer vision, such as Medical Image Segmentation. In conclusion, our major contribution is to propose a network model optimization method of thin branch structure, design the TbsNet network architecture, and achieve excellent performance improvement.

Although TbsNet has excellent performance, the number of FLOPs is still large compared with other lightweight networks such as DenseNet (*Huang et al., 2017*), MobileNet (*Sandler et al., 2018*), *etc*. In future work, we will focus on reducing the FLOPs and inference time to continuously improve TbsNet and apply it to more downstream tasks in computer vision.

| Depth (Layers) | Width | Set_group (Light) | Shortcut (Kernel) | Params (M) | | FLOPs (G) | | Top 1 Acc CIFAR-10 (%) | | Top 1 Acc CIFAR-100 (%) | |
|---|---|---|---|---|---|---|---|---|---|---|---|
| | | | | $\beta = -1$ | $\beta = 1$ | $\beta = -1$ | $\beta = 1$ | $\beta = -1$ | $\beta = 1$ | $\beta = -1$ | $\beta = 1$ |
| 17 | $w_1$ | × | 3 | 6.31 | 9.27 | 23.46 | 35.12 | 96 | 95.92 | 80.19 | 81.03 |
| | $w_1$ | × | 1 | 4.06 | 7.02 | 16.57 | 28.23 | 95.83 | 95.92 | | |
| | $w_1$ | √ | 3 | 3.65 | 4.16 | 12.42 | 14.43 | 95.49 | 95.66 | 78.31 | 79.25 |
| | $w_1$ | √ | 1 | 1.4 | 1.91 | 5.53 | 7.54 | 94.93 | 95.26 | | |
| | $w_2$ | × | 3 | 9.24 | 15.95 | 35.91 | 56.49 | 96.28 | 96.48 | 80.82 | 81.72 |
| | $w_2$ | × | 1 | 5.93 | 12.64 | 25.76 | 46.34 | 95.87 | 96.02 | | |
| | $w_2$ | √ | 3 | 5.46 | 6.59 | 18.72 | 22.22 | 95.81 | 95.91 | 79.82 | 80.22 |
| | $w_2$ | √ | 1 | 2.15 | 3.27 | 8.57 | 12.07 | 95.43 | 95.66 | | |
| | $w_3$ | × | 3 | 14.23 | 26.18 | 41.04 | 68.66 | 96.3 | 96.38 | 81.18 | 81.91 |
| | $w_3$ | × | 1 | 8.91 | 20.87 | 28.79 | 56.41 | 95.79 | 96.18 | | |
| | $w_3$ | √ | 3 | 8.64 | 10.63 | 22.09 | 26.74 | 95.91 | 96.14 | 79.95 | 80.86 |
| | $w_3$ | √ | 1 | 3.33 | 5.32 | 9.84 | 14.49 | 95.46 | 95.81 | | |
| | $w_4$ | × | 3 | 24.84 | 48.82 | 49.10 | 76.70 | 95.98 | 96.11 | 81.24 | 81.35 |
| | $w_4$ | × | 1 | 15.53 | 39.51 | 33.79 | 61.38 | 95.71 | 95.7 | | |
| | $w_4$ | √ | 3 | 16.62 | 23.87 | 28.14 | 36.82 | 95.92 | 96.01 | 80.81 | 81.71 |
| | $w_4$ | √ | 1 | 7.31 | 14.56 | 12.83 | 21.51 | 95.28 | 95.9 | | |
| 23 | $w_1$ | × | 3 | 6.89 | 9.84 | 34.82 | 46.48 | 96.19 | 96.31 | 81.57 | 81.52 |
| | $w_1$ | × | 1 | 4.64 | 7.59 | 27.93 | 39.59 | 96.21 | 96.48 | | |
| | $w_1$ | √ | 3 | 3.79 | 4.3 | 15.39 | 17.39 | 95.91 | 96.01 | 79.22 | 80.31 |
| | $w_1$ | √ | 1 | 1.54 | 2.05 | 8.49 | 10.50 | 95.77 | 95.95 | | |
| | $w_2$ | × | 3 | 10.27 | 16.97 | 56.03 | 76.61 | 96.55 | 96.75 | 82.17 | 82.28 |
| | $w_2$ | × | 1 | 6.95 | 13.66 | 45.89 | 66.47 | 96 | 96.49 | | |
| | $w_2$ | √ | 3 | 5.72 | 6.85 | 23.91 | 27.41 | 96.07 | 96.21 | 80.69 | 81.23 |
| | $w_2$ | √ | 1 | 2.41 | 3.53 | 13.77 | 17.26 | 95.85 | 96.19 | | |
| | $w_3$ | × | 3 | 15.25 | 27.21 | 61.17 | 88.78 | 96.6 | 96.68 | 81.88 | 82.08 |
| | $w_3$ | × | 1 | 9.94 | 21.89 | 48.92 | 76.53 | 96.24 | 96.4 | | |
| | $w_3$ | √ | 3 | 8.9 | 10.89 | 27.29 | 31.94 | 96.37 | 96.1 | 81.44 | 82.07 |
| | $w_3$ | √ | 1 | 3.58 | 5.58 | 15.04 | 19.69 | 95.82 | 96.27 | | |
| | $w_4$ | × | 3 | 25.86 | 49.84 | 69.22 | 96.82 | 96.46 | 96.46 | 81.94 | 81.55 |
| | $w_4$ | × | 1 | 16.55 | 40.53 | 53.91 | 81.50 | 95.9 | 96.12 | | |
| | $w_4$ | √ | 3 | 16.88 | 24.13 | 33.33 | 42.01 | 96.16 | 96.31 | 81.06 | 81.71 |
| | $w_4$ | √ | 1 | 7.57 | 14.82 | 18.02 | 26.70 | 95.86 | 96.13 | | |
| 29 | $w_1$ | × | 3 | 7.47 | 10.42 | 46.18 | 57.84 | 96.58 | 96.73 | 81.71 | 82.43 |
| | $w_1$ | × | 1 | 5.22 | 8.17 | 39.29 | 50.95 | | | | |
| | $w_1$ | √ | 3 | 3.94 | 4.45 | 18.35 | 20.35 | 96.14 | 96.12 | 80.86 | 80.37 |
| | $w_1$ | √ | 1 | 1.69 | 2.2 | 11.46 | 13.46 | | | | |
| | $w_2$ | × | 3 | 11.29 | 17.99 | 76.15 | 96.73 | 96.58 | 96.66 | 82.63 | 82.67 |

(Continued)

| Appendix 1 (continued) | | | | | | | | | | | |
|---|---|---|---|---|---|---|---|---|---|---|---|
| **Depth (Layers)** | **Width** | **Set_group (Light)** | **Shortcut (Kernel)** | **Params (M)** | | **FLOPs (G)** | | **Top 1 Acc CIFAR-10 (%)** | | **Top 1 Acc CIFAR-100 (%)** | |
| | | | | β = −1 | β = 1 | β = −1 | β = 1 | β = −1 | β = 1 | β = −1 | β = 1 |
| | $w_2$ | × | 1 | 7.97 | 14.68 | 66.01 | 86.59 | | | | |
| | $w_2$ | √ | 3 | 5.98 | 7.11 | 29.10 | 32.60 | 96.27 | 96.39 | 81.35 | 81.9 |
| | $w_2$ | √ | 1 | 2.67 | 3.79 | 18.96 | 22.45 | | | | |
| | $w_3$ | × | 3 | 16.27 | 28.23 | 81.29 | 108.91 | 96.6 | 96.79 | 82.9 | 82.92 |
| | $w_3$ | × | 1 | 10.96 | 22.91 | 69.04 | 96.66 | | | | |
| | $w_3$ | √ | 3 | 9.16 | 11.15 | 32.48 | 37.13 | 96.55 | 96.55 | 81.8 | 82.7 |
| | $w_3$ | √ | 1 | 3.84 | 5.84 | 20.23 | 24.88 | | | | |
| | $w_4$ | × | 3 | 26.88 | 50.86 | 89.34 | 116.94 | 96.38 | 96.64 | 82.8 | 82.72 |
| | $w_4$ | × | 1 | 17.57 | 41.55 | 74.03 | 101.63 | | | | |
| | $w_4$ | √ | 3 | 17.14 | 24.39 | 38.53 | 47.20 | 96.4 | 96.47 | 82.36 | 82.27 |
| | $w_4$ | √ | 1 | 7.83 | 15.08 | 23.21 | 31.89 | | | | |
| 35 | $w_1$ | × | 3 | 8.38 | 11.34 | 57.48 | 69.14 | 96.64 | 96.81 | 82.51 | 82.26 |
| | $w_1$ | × | 1 | 6.13 | 9.09 | 50.59 | 62.25 | | | | |
| | $w_1$ | √ | 3 | 4.17 | 4.68 | 21.25 | 23.26 | 96.29 | 96.34 | 81.05 | 81.6 |
| | $w_1$ | √ | 1 | 1.92 | 2.43 | 14.36 | 16.37 | | | | |
| | $w_2$ | × | 3 | 12.92 | 19.62 | 96.20 | 116.78 | 96.75 | 96.77 | 82.86 | 83.31 |
| | $w_2$ | × | 1 | 9.61 | 16.31 | 86.06 | 106.64 | | | | |
| | $w_2$ | √ | 3 | 6.39 | 7.52 | 34.22 | 37.72 | 96.43 | 96.65 | 82.16 | 82.49 |
| | $w_2$ | √ | 1 | 3.08 | 4.21 | 24.08 | 27.57 | | | | |
| | $w_3$ | × | 3 | 17.9 | 29.86 | 101.34 | 128.96 | 96.7 | 96.86 | 83.31 | 83.29 |
| | $w_3$ | × | 1 | 12.59 | 24.55 | 89.09 | 116.71 | | | | |
| | $w_3$ | √ | 3 | 9.57 | 11.57 | 37.60 | 42.25 | 96.59 | 96.73 | 82.53 | 82.59 |
| | $w_3$ | √ | 1 | 4.26 | 6.25 | 25.35 | 30.00 | | | | |
| | $w_4$ | × | 3 | 28.51 | 52.49 | 109.39 | 164.58 | 96.71 | 96.71 | 83.56 | 83.53 |
| | $w_4$ | × | 1 | 19.2 | 43.18 | 94.08 | 149.27 | | | | |
| | $w_4$ | √ | 3 | 17.56 | 24.81 | 43.65 | 52.32 | 96.81 | 96.88 | 82.62 | 82.88 |
| | $w_4$ | √ | 1 | 8.24 | 15.49 | 28.33 | 37.01 | | | | |
| 44 | $w_1$ | × | 3 | 9.76 | 12.72 | 74.44 | 86.10 | 96.64 | 96.69 | 82.04 | 82.56 |
| | $w_1$ | × | 1 | 7.51 | 10.47 | 67.55 | 79.21 | | | | |
| | $w_1$ | √ | 3 | 4.53 | 5.04 | 25.61 | 27.62 | 96.34 | 96.4 | 81.43 | 82.29 |
| | $w_1$ | √ | 1 | 2.28 | 2.79 | 18.72 | 20.73 | | | | |
| | $w_2$ | × | 3 | 15.37 | 22.07 | 126.28 | 146.86 | 96.81 | 97.02 | 83.04 | 82.99 |
| | $w_2$ | × | 1 | 12.05 | 18.76 | 116.13 | 136.71 | | | | |
| | $w_2$ | √ | 3 | 7.01 | 8.14 | 41.90 | 45.40 | 96.62 | 96.67 | 82.41 | 82.78 |
| | $w_2$ | √ | 1 | 3.7 | 4.83 | 31.76 | 35.25 | | | | |
| | $w_3$ | × | 3 | 20.35 | 32.31 | 131.41 | 159.03 | 96.94 | 96.81 | 82.94 | 82.51 |
| | $w_3$ | × | 1 | 15.04 | 26.99 | 119.16 | 146.78 | | | | |

**Appendix 1** (*continued*)

| Depth (Layers) | Width | Set_group (Light) | Shortcut (Kernel) | Params (M) | | FLOPs (G) | | Top 1 Acc CIFAR-10 (%) | | Top 1 Acc CIFAR-100 (%) | |
|---|---|---|---|---|---|---|---|---|---|---|---|
| | | | | $\beta = -1$ | $\beta = 1$ | $\beta = -1$ | $\beta = 1$ | $\beta = -1$ | $\beta = 1$ | $\beta = -1$ | $\beta = 1$ |
| | $w_3$ | √ | 3 | 10.19 | 12.19 | 45.28 | 49.93 | 96.83 | 96.93 | 82.93 | 83.53 |
| | $w_3$ | √ | 1 | 4.88 | 6.87 | 33.03 | 37.68 | | | | |
| | $w_4$ | × | 3 | 30.96 | 54.94 | 139.47 | 167.06 | 96.57 | 96.71 | 83.21 | 83.32 |
| | $w_4$ | × | 1 | 21.65 | 45.63 | 124.16 | 151.75 | | | | |
| | $w_4$ | √ | 3 | 18.18 | 25.43 | 51.33 | 60.00 | 96.61 | 96.62 | 82.86 | 83.24 |
| | $w_4$ | √ | 1 | 8.87 | 16.11 | 36.01 | 44.69 | | | | |

**Note:**
The params were calculated by torchsummary and the FLOPs were fvcore. Width$_{list}$ = {w1, w2, w3, w4} = {[96, 192, 384, 512], [128, 256, 384, 768], [128, 256, 512, 1,024], [128, 256, 512, 2,048]}. The shape of the input feature map was [1, 3, 224, 224]. Expansion ratio $\beta = [-1, 1]$.

### Funding

This work was supported by the First Batch of Industry-University Cooperation Collaborative Education Projects in 2021 (No. 202101202002), the Natural Science Foundation of Colleges and Universities of Anhui Province (No. KJ2020A0773) and the Excellent top-of-the-line Talent Training Program of Anhui Province Colleges and Universities (No. gxgnfx2019063). The funders had no role in study design, data collection and analysis, decision to publish, or preparation of the manuscript.

### Grant Disclosures

The following grant information was disclosed by the authors:
First Batch of Industry-University Cooperation Collaborative Education Projects in 2021: 202101202002.
Natural Science Foundation of Colleges and Universities of Anhui Province: KJ2020A0773.
Excellent top-of-the-line Talent Training Program of Anhui Province Colleges and Universities: gxgnfx2019063.

### Competing Interests

The authors declare that they have no competing interests.

### Author Contributions

- Xiujian Hu conceived and designed the experiments, performed the experiments, analyzed the data, performed the computation work, prepared figures and/or tables, authored or reviewed drafts of the article, and approved the final draft.
- Guanglei Sheng conceived and designed the experiments, analyzed the data, performed the computation work, prepared figures and/or tables, and approved the final draft.
- Piao Shi performed the experiments, authored or reviewed drafts of the article, and approved the final draft.

- Yuanyuan Ding analyzed the data, authored or reviewed drafts of the article, and approved the final draft.

## Data Availability

The data is available at the following links:

- Cifar10&100 dataset:

Hu, Xiujian (2022): cifar10&cifar100. figshare. Dataset. https://doi.org/10.6084/m9.figshare.21532920.v2.

- Tiny-Imagenet dataset:

Hu, Xiujian (2023): Tiny_Imagenet. figshare. Dataset. https://doi.org/10.6084/m9.figshare.22012529.v1.

- Synapse2015 dataset:

Hu, Xiujian (2023): Synapse2015. figshare. Dataset. https://doi.org/10.6084/m9.figshare.22012538.v1.

The data are also available at:

- University of Toronto: CIFAR-10 and Cifar100 dataset:

https://www.cs.toronto.edu/~kriz/cifar.html.

- GitHub: Tiny-ImageNet dataset:

https://github.com/seshuad/IMagenet.

- Synapse:

https://www.synapse.org/#!Synapse:syn3193805/files/.

- Image Net: ImageNet-1K dataset:

https://image-net.org/.

## Supplemental Information

Supplemental information for this article can be found online at http://dx.doi.org/10.7717/peerj-cs.1429#supplemental-information.

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
