# Peer review of "TbsNet: the importance of thin-branch structures in CNNs"

_PeerJ Computer Science, doi:10.7717/peerj-cs.1429_

## Round 0.1 · original submission · Major Revisions

Please revise your manuscript according to the obtained comments.

Reviewer 1 ·

Basic reporting

See the Additional comments section.

Experimental design

See the Additional comments section.

Validity of the findings

See the Additional comments section.

Additional comments

In this study, the authors analyzed the influence of some factors on network performance, proposed some strategies for constructing an efficient convolution network, and designed a convolution neural network named TbsNet.


1. The first paragraph of the abstract should have been more clearly stated.
2. The reference to UNet in line 35 is not necessary.
3. In line 71, it should read "trade-offs in terms of depth and width of the network".
4. The features of the designed architecture should be fully described in the article, not only the TbsNet architecture shown in the figure.
5. Line 354, "the TBS block retains the setting of groups as an option." should be changed to "the TBS block retains the setting of the number of groups as an option." 6. line 388, "the TBS block retains the setting of the number of groups as an option."
6. Line 388, "Best Accuracy on Width and Depth" should be changed to "The maximum Width and Depth".
7. Line 494 has a syntax problem, "the" should be added before "upsampling" and "downsampling".
8. The authors should explain why the inference time is not included in the network performance metrics.
9. The authors should explain the effect of feature map size on the scaling ability of the network.
10. The authors should explain why the experiments were designed and the logical relationship between the experiments.
11. The authors should explain the logic between the chosen dataset and the experiments.
12. The format of the references should be further modified.

Reviewer 2 ·

Basic reporting

The article must be written in English and must use clear, unambiguous, technically correct text. The article must conform to professional standards of courtesy and expression.

Experimental design

no comment

Validity of the findings

no comment

Additional comments

1. The abstract does not clearly state the problems, methods, and results. Therefore, further modification is required.
2. You are advised to explain why the inference time and memory usage are not selected as network performance indicators.
3. In the section "Architecture", the "Thin" feature of TbsNet's overall architecture should first be highlighted.
4. It is recommended to add NAS to the TbsNet network architecture in future work.
5. It is recommended to explain the relationship between the data set used and the experimental objectives.
6. It is suggested to explain why the ES (Compression-expansion) operation is added, what effect it has on network performance, and draw conclusions.
7. It is suggested to further study ERF (effective receptive field) in future work.
8. The language expression of the manuscript needs further calibration.

Reviewer 3 ·

Basic reporting

The authors presented the TbsNet architecture in CNN. Overall article has merit to be published. However, authors need to improve the quality of the manuscript to enhance the readability of the manuscript.

Experimental design

1. More explanation is required on the selection of Params, FLOPs, and Accuracy as network performance indicators.
2. It is necessary to explain why medical image segmentation is chosen as the application case of TbsNet in downstream tasks of computer vision.
3. The adaptive capability of TbsNet's variant structure needs to be explained. It is too brief in the manuscript

Validity of the findings

1. It is necessary to explain the consistency between the selected data set and the experimental objective.
2. The relationship between feature map size and network scaling capability needs to be further explained.
3. It is suggested that more studies should be conducted on how to increase the effective receptive field size in future work.

Additional comments

The format of the references needs to be modified

---

## Round 0.2 · Minor Revisions

Please further revise your paper according to the reviewers' comments.

Reviewer 1 ·

Basic reporting

See the additional comments.

Experimental design

See the additional comments.

Validity of the findings

See the additional comments.

Additional comments

The authors addressed my comments carefully. Now I only have some minor comments.

1. The reference to Appendix 1 in line 269 should be included as part of the paper. 2.
2. In line 427, the title "Benchmarking" should be moved to the next line.
3. In line 189, the strategy for building convolutional network architectures should be discussed in more detail.
4. The summary of the proposed approach should be strengthened in the conclusion.
5. The presentation of the dataset used should be streamlined.

Reviewer 3 ·

Basic reporting

Authors have made significant improvements on the manuscript. However, minor issues need to be addressed to enhance the quality of the manuscript.

1. Appendix 1 mentioned in line 269 should be included as part of the paper content.
2. The title "Benchmarking" should be moved to the next line in line 427.

Experimental design

1. In line 189, the strategy for building convolutional network architecture should be discussed in more detail.

2. The description of the used data set should be simplified.

Validity of the findings

In the conclusion, a stronger summary of the proposed methods should be provided.

---

## Round 0.3 · accepted · Accept

The revised paper has satisfactorily addressed all the reviewers' concerns and suggestions. I suggest to accept this manuscript.

Reviewer 2 ·

Basic reporting

no comment

Experimental design

no comment

Validity of the findings

no comment

Additional comments

I think this manuscript is okay.

Reviewer 3 ·

Basic reporting

Authors have addressed all the comments

Experimental design

Authors have addressed all the comments

Validity of the findings

Authors have addressed all the comments